

# A journey through the *Corynebacterium pseudotuberculosis* proteome promotes insights into its functional genome

Wanderson Marques da Silva[1], Nubia Seyffert[2], Artur Silva[3] and Vasco Azevedo[4]

[1] Institute of Agrobiotechnology and Molecular Biology-(INTA/CONICET), Hurlingham, Buenos Aires, Argentina
[2] Institute of Health Sciences, Federal University of Bahia, Salvador, Bahia, Brazil
[3] Laboratory of Genomics and Bioinformatics, Center of Genomics and Systems Biology, Institute of Biological Sciences, Federal University of Para, Belém, Pará, Brazil
[4] Genetics, Ecology and Evolution, Federal University of Minas Gerais, Belo Horizonte, Minas Gerais, Brazil

## ABSTRACT

**Background:** *Corynebacterium pseudotuberculosis* is a Gram-positive facultative intracellular pathogen and the etiologic agent of illnesses like caseous lymphadenitis in small ruminants, mastitis in dairy cattle, ulcerative lymphangitis in equines, and oedematous skin disease in buffalos. With the growing advance in high-throughput technologies, genomic studies have been carried out to explore the molecular basis of its virulence and pathogenicity. However, data large-scale functional genomics studies are necessary to complement genomics data and better understating the molecular basis of a given organism. Here we summarize, MS-based proteomics techniques and bioinformatics tools incorporated in genomic functional studies of *C. pseudotuberculosis* to discover the different patterns of protein modulation under distinct environmental conditions, and antigenic and drugs targets.
**Methodology:** In this study we performed an extensive search in Web of Science of original and relevant articles related to methods, strategy, technology, approaches, and bioinformatics tools focused on the functional study of the genome of *C. pseudotuberculosis* at the protein level.
**Results:** Here, we highlight the use of proteomics for understating several aspects of the physiology and pathogenesis of *C. pseudotuberculosis* at the protein level. The implementation and use of protocols, strategies, and proteomics approach to characterize the different subcellular fractions of the proteome of this pathogen. In addition, we have discussed the immunoproteomics, immunoinformatics and genetic tools employed to identify targets for immunoassays, drugs, and vaccines against *C. pseudotuberculosis* infection.
**Conclusion:** In this review, we showed that the combination of proteomics and bioinformatics studies is a suitable strategy to elucidate the functional aspects of the *C. pseudotuberculosis* genome. Together, all information generated from these proteomics studies allowed expanding our knowledge about factors related to the pathophysiology of this pathogen.

Corresponding author
Vasco Azevedo,
vascoariston@gmail.com

## INTRODUCTION

*Corynebacterium pseudotuberculosis* is a Gram-positive facultative intracellular pathogen belonging to the CMNR (*Corynebacterium-Mycobacterium-Nocardia-Rhodococcus*) group (*Barksdale, 1970*). This supra-generic group of *Actinomycetes* exhibits a specific cell wall composed of peptidoglycan, arabinogalactan, and mycolic acids and a high chromosomal G + C content (*Burkovski, 2013*; *Ventura et al., 2007*). *C. pseudotuberculosis* may be distinguished into two biovars according to nitrate reductase production: biovar *ovis* (nitrate negative) and biovar *equi* (nitrate positive) (*Biberstein, Knight & Jang, 1971*; *Dorella et al., 2006a*). Biovar *ovis* strains cause caseous lymphadenitis in small ruminants (*Williamson, 2001*; *Baird & Fontaine, 2007*) and mastitis mainly in dairy cattle (*Shpigel et al., 1993*). On the other hand, biovar *equi* strains cause ulcerative lymphangitis (*Britz et al., 2014*; *Haas et al., 2017*) and abscesses in the internal organs of the equines and oedematous skin disease in buffalos (*Selim, 2001*).

The infection by this pathogen is globally reported and causes significant economic losses affecting meat, wool, and milk production (*Haas et al., 2017*; *Kumar et al., 2012*; *Windsor, 2011*). The pathogenic process of *C. pseudotuberculosis* begins with colonization and replication within lymph nodes in the initial site of infection. In this initial step occurs the pyogranuloma formation. Due to its ability to survive and multiply within the macrophage phagosome, this pathogen begins a new cycle of bacterial replication. In this second cycle occurs death of the host cell and subsequent release and dissemination of bacterial *via* the lymphatic or circulatory system, allowing the pathogen to infect visceral organs and lymph nodes, where it ultimately induces lesion formation (*Batey, 1986*; *Pépin et al., 1994*; *Stefańska et al., 2010*).

In regard to virulence factors that contribute to *C. pseudotuberculosis* pathogenesis, phospholipase D (PLD) is described as major virulence factor of this bacterium which is involved in the dissemination of *C. pseudotuberculosis* from the site initial of infection to the lymph nodes (*McNamara, Bradley & Songer, 1994*) and reduction of the viability ovine neutrophils (*Yozwiak & Songer, 1993*) and macrophage (*McKean, Davies & Moore, 2007*). In addition, other determinants of virulence have been described as *fagABCD* operon (*Billington et al., 2002*) and CiuA siderophore (*Ribeiro et al., 2014*), both associated to iron uptake, cell wall components (*Muckle & Gyles, 1983*), CP40 serine protease (*Walker et al., 1994*; *Wilson, Brandon & Walker, 1995*), PhoP regulator of the PhoPR system two-component system (*Tiwari et al., 2014*), and OppD related to adherence and infection macrophages (*Moraes et al., 2014*).

The availability of whole-genome sequences of several *C. pseudotuberculosis* strains has provided information about the molecular basis on its dynamic physiology as well as pathogenesis. Upon the genome sequencing of FCR41 strain, *Trost et al. (2010)* proposed the following genes like putative virulence factors: *nanH* (neuraminidase H), *rpfI* (resuscitation-promoting factor interacting protein), *rpfA*

and *rpfB* (resuscitation-promoting factors A and B), *nor* (nitric oxide reductase), *dtsR1* (acetyl-CoA carboxylase β-subunit involved in fatty acid synthesis), *dtsR2* and *accD3* (acetyl-CoA carboxylase, beta subunit, involved in the biosynthesis of mycolic acid) and *spaC* (adhsin component of adhesin) and *nrpS1* and *nrpS2* (nonribosomal peptide synthetase 1 and 2). Currently, 124 genomes of *C. pseudotuberculosis* are publicly available at NCBI repositories. A number of these genomic studies have been carried out by our research group. These collections of genome data have been demonstrated across several *C. pseudotuberculosis*, presenting an average genome size of approximately 2.3 Mb, with a core-genome having approximately 768 genes and 2,795 belonging to the accessory genome (*Araújo et al., 2019*). In addition a pan-genomic analysis showed that pathogenic corynebacteria exhibited a close relationship and a clonal-like behavior among isolates of *C. pseudotuberculosis* (*Soares et al., 2013a*). On the other hand, through these genomic data, the *C. pseudotuberculosis* proteome was predicted with an average of approximately 2,098 proteins (*Ruiz et al., 2011*; *Santos et al., 2012*).

Proteomic study promotes insights about the functional state of the genome at the protein level; in microbiology, proteomics studies have contributed to broadening our knowledge about microbial adaptation in different types of environmental stress, pathogenesis, host-pathogen interaction, and microbial meta-proteomics (*Cordwell, Nouwens & Walsh, 2001*; *Schmidt & Völker, 2011*; *Otto et al., 2012*; *Broadbent et al., 2016*). In this context, proteomics studies have been performing aiming to characterize the *C. pseudotuberculosis* proteome (Table 1). The first studies of the proteome of *C. pseudotuberculosis* were based on one-dimensional sodium dodecyl sulphate polyacrylamide gel electrophoresis (SDS-PAGE) and immunoblotting (*Ellis et al., 1991*; *Muckle et al., 1992*; *Braithwaite et al., 1993*). The main objective of these studies is to describe the protein band patterns expressed by various strains and identify targets for the development of immunodiagnostics and vaccine targets to combat diseases caused by this pathogen. Mass spectrometry (MS)-based proteomics promotes a high level of detection and identification of proteins. Currently, researchers employ techniques such as two-dimensional gel electrophoresis (2-DE) and MALDI-TOF, liquid chromatography coupled to mass spectrometry (LC-MS), and multidimensional protein identification technology (MudPIT) that combines several two-dimensional chromatographic methodologies and MS to characterize and quantify the proteome of *C. pseudotuberculosis*.

Once, the last work which provides information about proteomic studies carried out with *C. pseudotuberculosis* was written by *Dorella et al. (2013)*. This work aims to promote an update regarding the latest advances obtained on the functional genome of *C. pseudotuberculosis* through proteomic studies. In addition, this study will provide an overview of immunoproteomics, bioinformatic, and a genetic tool used to explore the proteome of this pathogen to identify potential targets for the development of vaccines and immunodiagnostic methods against illnesses caused by this pathogen. This review will surely interest other research teams working on other pathogens. Furthermore, our findings are of considerable interest to those who work on the identification of bacterial antigenic targets, pathogenesis, functional genomics, and the adaptive strategies of pathogenic bacteria.

**Table 1 List of the studies that contributed to characterization of the *C. pseudotuberculosis* proteome.**

| Proteomic technique/ approach | Strain/biovar | Sample | Main findings | Reference |
|---|---|---|---|---|
| SDS-PAGE and immunoblot | ATCC 19410_ovis | Whole-cell extract and culture filtrate | Description of protein band patterns produced *C. pseudotuberculosis* and recognized of protein by antibodies of naturally infected sheep. | *Ellis et al. (1991)* |
| SDS-PAGE and immunoblot | ATCC 19410_ovis | Whole-cell extract | Identification of *C. pseudotuberculosis* antigens using serum of naturally infected sheep and goats. | *Muckle et al. (1992)* |
| SDS-PAGE and immunoblot | ATCC 19410_ovis, ATCC 809_ovis, ATCC 43927_ovis, ATCC 43926_ovis, ATCC 43924_equi, ATCC 43925_equi, ATCC PVAMU101_equi | Extracellular proteins and Whole cells extract | Description of protein band patterns produced by various *C. pseudotuberculosis* strains. In addition to the recognition of protein by antibodies of naturally infected goats. | *Braithwaite et al. (1993)* |
| SDS-PAGE | T1_ovis, T2_ovis and 1002_ovis | Extracellular proteins | A CDM standarized to the growth of *C. pseudotuberculosis*. | *Moura-Costa et al. (2002)* |
| SDS-PAGE and immunoblot | T1_ovis and 1002_ovis | Extracellular proteins | The TPP method showed to be suitable to obtain secreted proteins of *C. pseudotuberculosis*. | *Paule et al. (2004b)* |
| LC-MS$^E$ | 1002_ovis and C231_ovis | Extracellular proteins | TPP/LC-MSE is a suitable approach to characterize extracellular proteins of *C. pseudotuberculosis*, and comparative analysis reveals the production of PLD, CP40, and CiuA only in C231_ovis. | *Pacheco et al. (2011)* |
| SERPA (2-DE MALDI-TOF/ TOF-MS/MS) | 1002_ovis | Extracellular proteins | Identification of six immunoreactive exoproteins (RpfB, NlpC/P60, efflux system protein, SlpA, and one unknown function protein). | *Seyffert et al. (2011)* |
| LC-MS$^E$ | 1002_ovis and ΔsigE | Extracellular proteins | Characterization of the set of extracellular proteins regulated by *sigE* during nitric oxide exposition. | *Pacheco et al. (2012)* |
| 2-DE MALDI-TOF/TOF-MS/MS | 1002_ovis and C231_ovis | Extracellular proteins | Detection of PLD only in C231_ovis and identification of 11 new extracellular proteins of *C. pseudotuberculosis*. | *Silva et al. (2013a)* |
| 2D-DIGE MALDI-TOF/TOF-MS/MS | 1002_ovis and C231_ovis | Extracellular proteins | Quantitative analysis reveals the modulation of proteins involved in the cell envelope, respiratory metabolism, and proteolysis. | *Silva et al. (2013b)* |
| SERPA (2-DE/LC-MS/MS) | 1002_ovis and C231_ovis | Extracellular proteins | Identification of immune-reactive extracellular proteins | *Seyffert et al. (2014)* |
| LC-MS$^E$/2-DE MALDI-TOF/TOF-MS/MS | 1002_ovis | Whole-cell extract | Network of proteins related to resistance and survival of *C. pseudotuberculosis* to nitrosative stress. | *Silva et al. (2014)* |
| LC-MS/MS | C231_ovis and RLC001_ovis | Surface proteins | Thirteen surface proteins of *C. pseudotuberculosis* were detected exclusively from bacterial shaving in caseous nodules. In addition, 49 host proteins also were detected in infected nodules. | *Rees et al. (2015a)* |
| Dimethylation labeling LC-MS/MS and label-free LC-MS/MS | C231_ovis, RLC001_ovis, RLC002_ovis and RLC003_ovis | Whole-cell extract | Proteins related to hypoxia, nutrient deficiency responses, and thiopeptide biosynthesis are more induced in field isolate than reference strain C231_ovis. | *Rees et al. (2015b)* |

| Proteomic technique/ approach | Strain/biovar | Sample | Main findings | Reference |
|---|---|---|---|---|
| LC-MS$^E$ | 1002_ovis | Extracellular proteins | Bacterial serial passage in a murine model induces the PLD and CP40 proteins. | Silva et al. (2017a) |
| LC-MS$^E$ | 258_equi | Extracellular ptoteins | Iron acquisition proteins are more induced in the exoproteome of 258_equi recovered from mice spleen. | Silva et al. (2017b) |
| LC-MS$^E$ | 1002_ovis and 258_equi | Whole-cell extract | Identification of protein strain-specific in 1002_ovis and 258_equi. | Silva et al. (2017c) |
| LC-MS/MS | VD57_ovis | Membrane-associated proteins | Bacterial grown in animal serum induce production of proteins associated with iron uptake and Opp system | Raynal et al. (2018) |
| LC-MS$^E$ | CAP3W_ovis and CAPJ4_ovis | Whole-cell extract | Proteins related to biological processes involved in biofilm formation, such as quorum sensing, metabolism, and cell wall components were more abundant in biofilm-forming strain. | De Sá et al. (2021) |

## SURVEY METHODOLOGY

Extensive literature research was conducted using the following electronic databases: Pubmed, Science Direct, Google Scholar, and Scopus. The keyword combinations such as: *C. pseudotuberculosis* and proteomic, *C. pseudotuberculosis* and proteome, *C. pseudotuberculosis* and bioinformatics, *C. pseudotuberculosis* and proteins, *C. pseudotuberculosis* and drugs, *C. pseudotuberculosis* and functional genomics, bacterial and exoproteome, bacterial and surfaceome, *C. pseudotuberculosis* and antigens, *C. pseudotuberculosis* and vaccine, bacterial and proteomic, *C. pseudotuberculosis* and genomics, *C. pseudotuberculosis* and hypothetical proteins, bacterial and moonlight proteins were utilized to build literature review. All articles were extensively studied to be utilized as references in the present review article.

### Strategies and proteomic approach utilized to characterize the *C. pseudotuberculosis* exoproteome

Bacteria can release proteins across external membranes to the extracellular milieu through different exportation and secretion pathways. Extracellular proteins are considered essential elements of physiology, pathogenicity, and virulence bacterial. These proteins are related to distinct processes such as adhesion and invasion to host cells, damage to host tissues, resistance to environmental stress, nutrient acquisition and subversion of the host's immune response (Desvaux et al., 2009; Galán & Waksman, 2018; Weber & Faris, 2018). On the other hand, some studies showed that exoproteins to be the potential target for the development of drugs (Maffei, Francetic & Subtil, 2017; Galán & Waksman, 2018). Thus, the characterization of the bacterial proteins present in the extracellular milieu, *i.e.*, exoproteome (Desvaux et al., 2009), besides promoting knowledge about

physiology and bacterial virulence, may represent targets important for drug and vaccine development as well as immunodiagnostic tests.

As described above, during the first studies, different groups used techniques as one-dimensional SDS-PAGE and immunoblot to characterize the extracellular proteins produced by *C. pseudotuberculosis* (*Ellis et al., 1991*; *Braithwaite et al., 1993*). These studies allowed the identification only of the molecular mass of the proteins. Moreover, these extracellular proteins come from bacterial growth in bacterial culture media such as brain-heart infusion (BHI) or Tryptone, which are culture media extremely rich in proteins. Then, to eliminate the presence of exogenous proteins from complex media in the bacterial culture supernatant, *Moura-Costa et al. (2002)* has developed a chemically-defined medium (CDM) for *C. pseudotuberculosis* culture. Interestingly, the bacterial growth in this medium allowed the production of extracellular proteins with high molecular weight (showed by SDS-PAGE) that are not produced when the bacterium was grown in the BHI medium (*Paule et al., 2004a*). Currently, no work has analyzed which components of this medium are responsible for inducing these proteins. Another study shows that this extracellular fraction from bacterial grown in CDM recognized the IgG of experimentally infected goats in an ELISA assay; in addition, the use of these extracellular proteins in an interferon-γ (IFN-γ) assay provided a more specific response when compared to the use of proteins obtained from bacterial whole-cell sonicated (*Paule et al., 2003*). These results demonstrate that in addition to being a medium is suitable for the growth of *C. pseudotuberculosis*; CDM induces antigenic proteins of this pathogen.

Subsequently, to optimize obtaining the extracellular proteins produced by *C. pseudotuberculosis*, a three-phase partitioning protocol (TPP) was standardized (*Paule et al., 2004b*). Studies have been demonstrated that the TPP method is a powerful technique for extraction, separation, and purification of bioactive molecules (*Dennison & Lovrien, 1997*; *Yan et al., 2018*; *Wang et al., 2019*). In the TPP method, the proteins are precipitated between the aqueous phase (upper phase) generated for the ammonium sulfate and the organic phase (lower phase), which contain *n*-butanol (*Dennison & Lovrien, 1997*). During the standardization of the TPP, *Paule et al. (2004b)* showed that lowering the pH of the culture medium supernatant before precipitation with 30% ammonium sulfate concentration promotes an increased qualitative extraction. This technique allowed the detection of extracellular immunoreactive proteins of low (16 kDa) and high (125 kDa) molecular weights not detected by other studies that used only the ammonium sulfate precipitation method (*Ellis et al., 1991*; *Braithwaite et al., 1993*; *Paule et al., 2003*). On the other hand, these precipitated proteins from the TPP technique were useful in detect sheep and goats with caseous lymphadenitis in an IFN-γ assay (*Rebouças et al., 2011*).

After the standardization of the CDM to *C. pseudotuberculosis* growth and TPP method to obtain extracellular proteins; the followings studies of the extracellular proteins of *C. pseudotuberculosis* combined CDM, TPP, and MS-based proteomics. These studies were based on both gel-based and gel-free techniques (Fig. 1). Gel-based analyses were carried out through 2-DE and 2D-DIGE techniques following mass spectrometry using

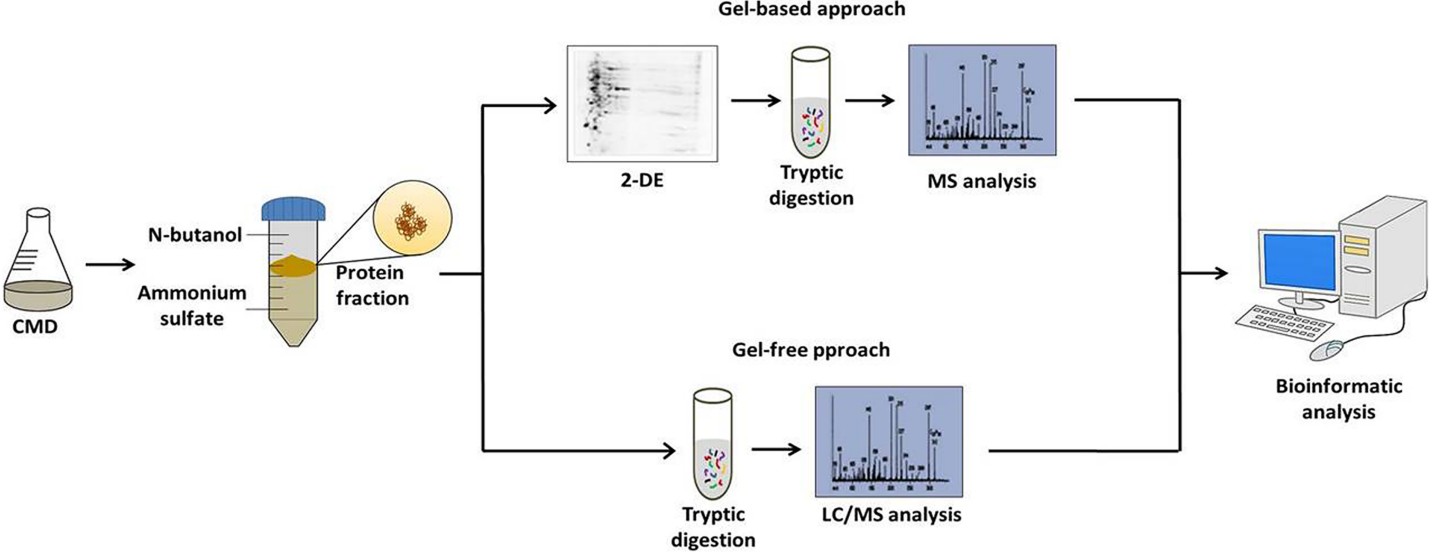

**Figure 1  Workflow of the strategy adopted to characterize the *C. pseudotuberculosis* exoproteome.**

MALDI-TOF/TOF-MS/MS (*Silva et al., 2013a*, *2013b*). However, to suppress some limitations of the gel-based analysis, such as sample amount, quantitative statistical power, and depth of proteome coverage; our group adopts the LC-MS$^E$ approach to characterize the exoproteome of *C. pseudotuberculosis* (*Pacheco et al., 2011*; *Pacheco et al., 2012*; *Silva et al., 2017a*; *Silva et al., 2017b*). This versatile high-throughput method based in label-free quantification has the advantage of having a high sensitivity, which makes it possible to increase the coverage level of the proteome (*Silva et al., 2006*). To complement these proteomic analyses, *Santos et al. (2012)* established a pipeline for *in silico* prediction of the subcellular localization of the *C. pseudotuberculosis* proteome. This pipeline includes the following software packages: (i) SurfG+ (*Barinov et al., 2009*) being the leading tool responsible for subcellular classifications in cytoplasmic (CYT), membrane (MEM), potentially surface exposed (PSE) and secreted (SEC), (ii) TatP 1.0 (*Bendtsen et al., 2005a*), (iii) SecretomeP 2.0, (*Bendtsen et al., 2005b*) and (iv) NclassG+ (*Restrepo-Montoya et al., 2011*) (Currently, NclassG+ tool is not available), with the latter three software's allowing the non-classical secretion prediction.

In a pioneer study, using TPP-LC/MS$^E$ approach (combination between TPP method and LC/MS$^E$ proteomic), *Pacheco et al. (2011)* carried out a comparative proteomic analysis between two *C. pseudotuberculosis* reference strains: low virulent strain 1002_*ovis* (isolated from goat in Brazil) (*Ribeiro et al., 1991*; *Meyer et al., 2002*) and high virulent strain C231_*ovis* (isolated from sheep in Australia) (*Simmons et al., 1998*) during exponential growth in CDM. This proteomic analysis identified 70 proteins in 1002_*ovis*, 67 proteins in C231_*ovis* and 44 proteins common between both strains. This study comparative allows the characterization of 93 different extracellular proteins of *C. pseudotuberculosis*, among them, 75% were predicted with exportation signals, showing the high capacity this approach adopted by the authors in identify exported proteins.

Comparative analysis identified virulence factors such as PLD exotoxin, CiuA siderophore, FagD siderophore, and Cp40 serine protease only in the extracellular proteome of C231_*ovis*. In turn, mainly proteases and unknown function proteins compose the exoproteome exclusive of 1002_*ovis*. Further, the following *moonlight* proteins: Elongation factor Tu, Elongation factor Ts, co-chaperonin GroES, Peroxiredoxin, and Phosphoglyceromutase are present in the exoproteome exclusive of this strain. According to the authors, the absence of PDL exotoxin in the extracellular proteome of the strain 1002_*ovis* would be related to its low virulence profile or the growth condition utilized in this study. A study showed that PLD induction occurs under specific environmental conditions (*McKean, Davies & Moore, 2007*).

*Silva et al. (2013a)* also observed the absence of both PLD exotoxin and CP40 in 1002_ovis exoproteome. In this study, the authors compared the exoproteome of 1002_*ovis* and C231_*ovis* during the stationary phase in CDM using 2-DE. Comparative analysis showed differences between the protein maps of the strains. After MALDI-TOF-MS/MS, 55 and 45 proteins were detected in 1002_*ovis* and C231_*ovis*, respectively. In the exclusive proteome of C231_*ovis* was detected the PLD exotoxin and CP40. In turn, in the exclusive exoproteome of 1002_*ovis* was detected the DsbG protein, a strain-specific protein. In addition, in this study, the authors reported 11 proteins (GroEL, DnaK, Elongation factor P, GAPDH, ABC-type transporter, Carbonic anhydrase, Manganese superoxide dismutase and three unknown function proteins) that were not previously detected in the study conducted by *Pacheco et al. (2011)*. Using the 2D-DIGE technique, *Silva et al. (2013b)* identified seventeen extracellular spots differentially produced between the 1002_*ovis* and C231_*ovis* (*Silva et al., 2013b*). These differentially induced proteins are related to energy production, cell wall, proteases, and unknown function proteins. Among them was detected the Lysozyme M1 that was not detected in other studies (*Pacheco et al., 2011*; *Silva et al., 2013a*). Together, these three proteomic studies (*Pacheco et al., 2011*; *Silva et al., 2013a*, *Silva et al., 2013b*) showed differences between the exoproteome of 1002_*ovis* and C231_*ovis*. In addition, these studies identified 105 different extracellular proteins of *C. pseudotuberculosis*. LC/MS[E] identified 57% (60 proteins) of the proteins due to the high sensibility of this method; 33% both gel-free and gel-based, and 12% only by gel-based approaches.

In another study, *Pacheco et al. (2012)* compared the exoproteomes of the *sigE* mutant and its parental strain 1002_*ovis* upon nitric oxide (NO) exposition. Various pathogens often find this reactive nitrogen species (RNS) inside macrophages (*Ferrari et al., 2011*). In this study, the authors showed that the sigma ($\sigma$) factor $\sigma^E$ is required in the resistance process of *C. pseudotuberculosis* to nitric oxide and peroxide stress (*Pacheco et al., 2012*). According to proteomic analysis the proteins thiol peroxidase, Glyoxalase/dioxygenase and ABC-type metal ion transport system are induced in wild-type strain upon NO-exposition. In the *sigE* mutant, the following proteins are in their exclusive exoproteome: four ABC-transporters, Alkyl hydroperoxide reductase subunit C, DnaK, Nitrite reductase periplasmic cytochrome c552, Periplasmic zinc-binding protein troA, Penicillin-binding protein A, Phosphocarrier protein HPr, Serine/threonine-protein kinase, Fructose-bisphosphate aldolase, and GAPDH.

The proteomic analysis of the culture supernatant of 1002_ovis (*Silva et al., 2017a*) and 258_*equi* (*Silva et al., 2017b*) strains isolated from mice spleen identified proteins that might play a role in the pathophysiology of *C. pseudotuberculosis*. In these studies, firstly mice have been experimentally infected and recovered strains (36h after infection) submitted to proteomic analysis. The proteomic screening in 1002_*ovis* was carried out by combining the LC-MS$^E$ and 2-DE/MALDI-TOF/TOF-MS/MS (*Silva et al., 2017a*). Interestingly, the virulence factors CP40 and PLD exotoxin are present in the exoproteome of recovered condition. These virulence factors had not been identified in previous studies of the exoproteome of this strain; when cultivated in laboratory standard conditions (*Pacheco et al., 2011*; *Silva et al., 2013a*; *Silva et al., 2013b*). This study showed that the induction of PLD exotoxin in strain 1002_*ovis* is influenced by environmental conditions. On the other hand, the 258_*equi* exoproteome was characterized utilizing only LC-MS$^E$. From this quantitative proteomic analysis, the more induced proteins after the recuperation process were related mainly to iron-acquisition such as CiuA, FhuD, FagC, HmuT, HmuV, and HtaA (*Silva et al., 2017b*). Studies have shown that CiuA (*Ribeiro et al., 2014*) and proteins from *fag* operon (*Billington et al., 2002*) are involved in the *C. pseudotuberculosis* pathogenesis. Moreover, several studies showed that proteins related to iron-uptake are required for bacterial pathogenesis (*Schaible & Kaufmann, 2004*; *Cassat & Skaar, 2013*; *Sheldon, Laakso & Heinrichs, 2016*). These results suggest that proteins related to iron-uptake can play an important role in the pathogenesis of 258_*equi*.

## Membrane-associated proteins of *C. pseudotuberculosis*

The bacterial cell envelope is a structure that corresponds to the cell surface, which protects bacterial cells and further mediates different types of interactions between bacteria and the environment (*Rajagopal & Walker, 2017*). In Gram-positive bacteria, the cell envelope is composed of two structures (i) cytoplasmic membrane (CM) and (ii) cell wall (CW). The CM is a selectively permeable barrier formed by a phospholipid bilayer composed of lipids, proteins, and carbohydrates. The proteins embedded in the CM can be either integral or anchored to the membrane. In turn, the CW is a structure that surrounds the cytoplasmic membrane and provides bacterial cell protection against mechanical stresses and osmotic lysis. The leading constituents of the CW are peptidoglycan, lipoteichoic acids, polysaccharides, glycopolymers, and proteins. The proteins found in the cell wall are covalently bound or bound through weak interactions (*Weidenmaier & Peschel, 2008*; *Rajagopal & Walker, 2017*; *Desvaux, Candela & Serror, 2018*). Thus, these sets of molecules that comprise the bacterial cell surface form the "bacterial surfaceome" and comprehensive analysis of the proteins present in the surfaceome is referred to as the "proteosurfaceome" (*Desvaux, Candela & Serror, 2018*). In bacterial pathogens, the surface proteins have a key role in the interplay between the bacteria and their host (*Kline et al., 2009*; *Hammerschmidt, Rohde & Preissner, 2019*).

Due to the importance of surface proteins in bacterial pathogenesis, *Rees et al. (2015a)* applied the shaving method to characterize the proteosurfaceome of a *C. pseudotuberculosis* isolate (RLC001 isolate) obtained directly from naturally infected sheep lymph nodes. Shaving is a powerful strategy to characterize surface-exposed

proteins. In this technique, the treatment of the intact bacterial cells with protease cleaves the proteins, and then the peptide fragments generated are submitted to MS analysis (*Rodríguez-Ortega et al., 2006*). In this study after MS analysis of the peptide fragments cleaved by trypsin, the authors characterized 89 proteins of *C. pseudotuberculosis* in caseous material related to different biological processes. Interestingly, when this set of proteins was compared with the proteomic profile of C231_*ovis* growth in culture media, the authors observed that the following proteins: chromosome partitioning protein ParB, Regulatory protein RecX, Protein FadF, Enolase, Fumarate hydratase class II, Argininosuccinate synthase, Sulfurtransferase, Glycerol-3-phosphate dehydrogenase, Aspartate-semialdehyde dehydrogenase, Lipid A biosynthesis lauroyl acyltransferase, Ascorbate-specific permease IIC component UlaA, Purine nucleoside phosphorylase DeoD-type, and Methylmalonyl-CoA carboxyltransferase 12S subunit were detected exclusively in the caseous material. In addition, they identified 49 host proteins from infected lymph nodes (Table S1), among them are important factors related to mainly immune response.

*Raynal et al. (2018)* characterized the surface proteome of *C. pseudotuberculosis* VD57_*ovis*, one virulent reference strain (*Moura-Costa et al., 2008*; *Almeida et al., 2016*). In this study, the authors utilized the following strategy; bacterial growth in culture medium composed by bovine fetal serum, obtaining of the proteins through an organic solvent enrichment protocol and LC-MS/MS analysis. Upon the MS analysis, the authors identified 116 proteins related to several biological processes. Among them are proteins involved in *C. pseudotuberculosis* pathogenesis, such as CiuA (*Ribeiro et al., 2014*) and FagA (*Billington et al., 2002*). Further are OppA4 and OppCD2 proteins of the operon *oppBCDA* (oligopeptide permease) involved in the adherence and infection macrophages (*Moraes et al., 2014*).

## Towards the whole-cell lysed proteome of *C. pseudotuberculosis*

The whole-cell proteomic strategy allows the identification and quantification of thousands of proteins. This strategy promotes a global proteomic analysis of a cell; due to the capacity to identify proteins from different subcellular fractions, which allows discovery and exploration of biological pathways. Thus studies applied the proteomic analysis of whole-cell protein extracts to access a high level of coverage of the global proteome of *C. pseudotuberculosis*.

*Rees et al. (2015b)* used both genomic and proteomic comparative analysis to identify the differences among three isolates of *C. pseudotuberculosis* obtained from lymph nodes of naturally infected sheep (field isolates) and reference strain C231_*ovis*. In this study, the authors utilized two MS-based quantification methods; (i) label-free and (ii) dimethylation labeling based on stable isotope. In this second method, the samples are labeled with heavy (C231_*ovis*) and light (field isolates) formaldehyde. From this comprehensive proteomic analysis 1.250 and 1.358 proteins were identified in label-free and dimethylation labeling methods, respectively. Although, genomic analysis reveals the presence of SNPs, and the gene presence or absence, among reference strain and field isolates. The proteins that have differential abundance were those conserved across all field

isolates and C231_*ovis*. Among the differentially induced proteins, eight proteins were more induced in all three field isolates (Pyruvate carboxylase, 5-formyltetrahydrofolate cyclo-ligase, Fructose-1,6-bisphosphatase, Precorrin-3B synthase, Citrate lyase subunit beta-like protein, uncharacterized protein YcaO, ABC transporter substrate-binding lipoprotein YvrC, and one hypothetical protein) and four were less induced in all three field isolates (Glutamate ABC transporter domain-containing ATP-binding protein, GluB, FhuD, and PstS). The use of strains recently isolated from natural hosts enabled access to proteins that might be required by this pathogen during the colonization processes.

The comparative analysis between the proteome of 1002_*ovis* and 258_*equi via* label-free proteomics characterized, 1.227 and 1.218 proteins para each strain, respectively (*Silva et al., 2017c*). Despite the quantitative differences observed in the core proteome such as induction of PLD in 258_*equi*, significant differences are present in the individual proteome of each strain, where strain-specific proteins were detected. The authors reported in the exclusive proteome of 1002_*ovis* a DNA methylase (ORF that codifies this protein is absent in the genome of 258_*equi*) and seven proteins involved in adhesion and motility cell, carbohydrate metabolism, lipid metabolism and unknown function that are annotated like pseudogene in the genome of strain 258_*equi*. On the other hand, in the exclusive proteome of 258_*equi* was detected one MoeB protein, two CRISPR-associated proteins, and three proteins with unknown function (Cp258_0076, Cp258_0585, and Cp258_0586) (ORF that codify these proteins is absent in the genome of 1002_*ovis*). In addition, 20 proteins present in the proteome of 258_*equi* related to amino acid metabolism, coenzyme metabolism, carbohydrate metabolism, energy metabolism, inorganic ion transport, post-translational modification, intracellular trafficking secretion and vesicular transport and unknown function are annotated like pseudogene in the genome of strain 1002_*ovis*. These differences observed reports information that can be useful to understand the aspects related to both pathogenesis and physiology of these strains.

Some study has been reported the biofilm formation in *C. pseudotuberculosis* (*Sá et al., 2013*; *Chandran et al., 2016*), this process offers protection during the bacterial growth in a hostile environment and contributes to the pathogen's persistence during the infection process (*Costerton, Stewart & Greenberg, 1999*). Studies shown that proteins associated to various biological processes are related to biofilm formation (*Rani & Babu, 2018*). In this context, *De Sá et al. (2021)* used label-free proteomics to compare the proteome of biofilm-forming (CAPJ4_*ovis*) and non-biofilm-forming (CAP3W_*ovis*) strains of *C. pseudotuberculosis* isolated from goats. The following proteins: GalT, GlpD, Zwf, and EtfA (metabolism), trehalose corynomycolyl transferase B, CwlM, and D-alanyl-Dalanine carboxypeptidase (cell wall components), and Phet (quorum sensing) were more induced in strain CAPJ4_*ovis*. These proteins are related to biological pathways required for biofilm formation in prokaryotic organisms/prokaryotes. Thus more studies are necessary to evaluate the role of these proteins in the biofilm formation in *C. pseudotuberculosis*.

In addition to these comparative proteomic studies between strains, the whole-cell extract of *C. pseudotuberculosis* has been characterized under stress conditions.

From label-free proteomic analysis using LC/MS$^E$ approach, *Silva et al. (2014)* reported the leading group of proteins utilized by strain 1002_*ovis* for resistance to oxide nitric. More induced proteins in stress condition were related to general stress response, cellular metabolism, detoxification, transcriptional regulation, and DNA synthesis and repair. Interestingly, the Glyoxalase/Bleomycin resistance protein/Dioxygenase involved in the detoxification process (*Thornalley, 2003*; *He & Moran, 2011*) was detected as more induced after stress exposition. The same protein was also detected in the study conducted by *Pacheco et al. (2012)*. According to the authors, the presence of this protein in both studies suggests that Glyoxalase/Bleomycin resistance protein/Dioxygenase can play an important physiological role during the *C. pseudotuberculosis* exposition to nitric oxide.

## Identification of antigenic proteins, drugs, and vaccine targets through the *C. pseudotuberculosis* proteome

Immunoproteomics includes the use of different proteomic technologies to identify antigenic peptides and proteins. In turn, immunoinformatics includes studies associated with immunological knowledge using bioinformatics software. Once cell surface-exposed and secreted proteins play an essential role in the host-pathogen interaction, these proteins represent a potential target in the formulation of drugs, biomarkers, and vaccines (*Vytvytska et al., 2002*; *Falisse-Poirrier et al., 2006*; *Fulton, Baltat & Twine, 2019*; *Raoufi et al., 2020*; *Oli et al., 2020*).

The Serological Proteome Analysis (SERPA) is a powerful immunoproteomic method often employed to identify immune-reactive proteins. In SERPA analysis, proteins are firstly resolved by 2-DE, then western blots are performed using reactive sera to detect the immunoreactive proteins, and finally, these proteins are identified by MS (*Klade, 2002*; *Fulton, Baltat & Twine, 2019*). Employing SERPA analysis, *Seyffert et al. (2011)* identified six immunoreactive exoproteins (RpfB, NlpC/P60, efflux system protein, SlpA, and one unknown function protein). This preliminary study combined the serum of goats infected with *C. pseudotuberculosis* and extracellular proteins of 1002_*ovis*. In a second study, *Seyffert et al. (2014)* utilized a pool of serum from goat and sheep infected by *C. pseudotuberculosis* and comparative proteomics to identify immune-reactive proteins. The comparative analysis between the exoproteome of 1002 and C231 allowed identifying 13 immunoreactive proteins non-redundant Interestingly, six immune-reactive exoproteins (Cphp1, Cphp2, Cphp3, Cphp4, Cphp6, and Cphp7) with unknown functions identified in the core-exoproteome are exclusive of *C. pseudotuberculosis*. This result is notary because these specific proteins represent the potential targets to be employed in diagnostic methods against caseous lymphadenitis (Syffert et al., 2014).

Using a reporter transposon-based system TnFuZ, *Dorella et al. (2006b)* identified 21 mutant strains with an alkaline phosphatase-positive activity that presented transposon insertions in genes that encode exported proteins. This system combines a reporter transposon-based system TnFuZ and the Tn*4001* transposable element with the alkaline phosphatase gene (*phoZ*) from *Enterococcus faecalis* (*Gibson & Caparon, 2002*). Posteriorly, the 21 mutant strains and their T1 parental strain have been employed in a
vaccination assay using a murine model (*Ribeiro et al., 2014*). In this study, mice have been intraperitoneally inoculated. Among all strains evaluated, the strain CZ171053 (a *ciuA* secreted-siderophore null presented) showed the best protection level (protection level, 80%) against the challenge with *C. pseudotuberculosis* MIC-6 a highly virulent wild strain. TnFuZ proved to be a rapid strategy in generating mutant strains (to gene encoding secreted proteins) for their posterior evaluation as candidate vaccines. *Galvão et al. (2017)* performed immunoscreening in a gene expression library of *C. pseudotuberculosis*. According to this analysis, the authors identified the following genes *dak2*, *fagA*, *fagB*, NlpC/P60 protein family, and LPxTG putative protein family, as potential targets to immunoassays or vaccine formulations, against caseous lymphadenitis.

The Mature Epitope Density (MED) is an immunoinformatics strategy extremely useful in selecting antigenic targets. This approach determines the epitopes concertation in exported proteins for major histocompatibility complex (MHC) I (*Santos et al., 2013*). The MED 1.0 Server pipeline (*Santos et al., 2013*) includes the following bioinformatics tools for predicting subcellular localization: SurfG+ 1.0 (*Barinov et al., 2009*) and TMHMM (*Krogh et al., 2001*). Using MED 1.0 Server, *Rezende et al. (2016)* identified three exoproteins: Cp1002_0126a, Cp1002_0369 (CP01850) and Cp1002_1957 (CP09720) as potential antigenic target. In immunoenzymatic assays, the authors showed that the rCP01850 recombinant protein presents high values of specificity and sensitivity, than rCP09720 recombinant protein. After this study, *Brum et al. (2017)* showed that mice immunized with the recombinant CP09720 (rCP09720) associated with aluminum hydroxide adjuvant present protection levels of 58.3%, against the challenged with MIC-6 virulent strain. In another study, the rCP01850 and rCP09720 recombinant proteins have been combined individually with recombinant PLD (rPLD) in a vaccination formulation (*Silva et al., 2018*). In this study, the rCP01850 + PLD formulation presented the best protection level in mice (subcutaneous inoculation route) against the challenge with strain MIC-6. On the other hand, when rCP09720 + rPLD was utilized in ELISA assay using serum samples from sheep with superficial lymph nodes abscesses, this combination demonstrated high sensitivity (97.5%) and specificity, than rCP01850 + PLD (*Silva et al., 2019*). More recently, a study showed that one vaccine formulation composed of rCP01850 protein and Brazilian red propolis as an adjuvant induces cellular and humoral immune responses (*Brilhante Bezerra et al., 2020*).

Reverse vaccinology is a concept adopted by the post-genomic era for vaccine targets discovery from genome sequence information of a pathogen (*Rappuoli, 2000*). However, due to the rapid availability of complete bacterial genomes, this concept has been expanded to pan-genomics reverse vaccinology (*Bambini & Rappuoli, 2009*). Using this concept, some studies have utilized reverse vaccinology as a strategy to identify potential vaccine targets inside the genome of *C. pseudotuberculosis*. The *in silico* prediction of the pan-exoproteome of five *C. pseudotuberculosis* strains (1002_*ovis*, C231_*ovis*, I19_*ovis*, FRC41_*ovis*, and PAT10_*ovis*) identified 306 exported proteins (122 predicted as SEC and 122 predicted as PSE proteins) common among all strains (*Santos et al., 2012*). This pioneering study allowed for the first time to build a dataset of possible vaccine targets. *Soares et al. (2013b)* carried out a screen in the genome of three *C. pseudotuberculosis*

strains 1002_*ovis*, 258_*equi*, and CIP 52.97_*equi*. In this study, secreted proteins, PSE proteins, or membrane proteins predicted by SurfG+ software (*Barinov et al., 2009*) were analyzed by the Vaxign software (*He, Xiang & Mobley, 2010*), resulting in a total of 49 core-proteins with antigenic properties. In another study, *Araújo et al. (2019)* carry out a pan-genomics reverse vaccinology study with sixty-five complete genomes of *C. pseudotuberculosis*. The core-proteome predicted was evaluated with Vaxign (*He, Xiang & Mobley, 2010*) and MED 1.0 Server pipeline (*Santos et al., 2013*). From these analyses, the following proteins: CopC, YkuE, NDH, MtrB, FtsI, and SenX3 were predicted as potential vaccine candidates by both reverse vaccinology strategies.

Druggability is a concept often utilized to identify putative therapeutic targets (*Owens, 2007*). In this context, two studies performed a structural druggability assessment of the *C. pseudotuberculosis* proteome (*Hassan et al., 2014*; *Radusky et al., 2015*). The modelomic analyses of the predicted proteomes of 15 strains of *C. pseudotuberculosis* led to the prediction of 34 potential targets, which are bacterial essentials proteins non-host homologous (human, horse, cow, and sheep). These proteins are related to both physiological processes and virulence and represent a dataset composed of potential druggable targets that may be in the development of compounds against *C. pseudotuberculosis* infection. In another study, *Barh et al. (2011)* identified 38 conserved common targets among *C. pseudotuberculosis* strains (1002_*ovis*, C231_*ovis*, I19_*ovis*, and FRC41_*ovis*) and CMN species (*C. diphtheriae*, *C. glutamicum*, *Mycobacterium tuberculosis*). In this study, the authors applied the comparative and subtractive genomics approaches to identify the potential targets. Among the 38 potential targets, 20 are non-homologous to goats, sheep, bovine, horses, and humans. Further, the modeled analysis and virtual screening to the targets MurA, MurE, FolP, NrdL, dcd, NrdH found five specific compounds for each target, in addition, one compound [c5( Dcd)/c1 (NrdL)] show specificity in targeting dcd and nrdL. On the other hand, the *in silico* prediction of the Protein-protein interaction networks of 15 *C. pseudotuberculosis* strains allow the identification of 41 essential proteins of this pathogen, non-host homologous, the authors suggest this set of proteins as potential therapeutic targets (*Folador et al., 2016*). Among them, 24 proteins had no significant hit against at least one host; the authors suggest this set of proteins as potential therapeutic targets.

Finally, when summarized all proteins identified across these different approaches described in this section, the following proteins: 50S ribosomal protein L1, 50S ribosomal protein L30, HlyD, HtaA domain-containing protein, NLP/P60 protein, RpfB, NrdI, SenX3, SlpA, NusG, Hypothetical protein (Cp1002_0126a/ ADL20032.2) and CmtB/ esterase (CP09720) were identified at least two different studies (Table S2). Until now, the following targets: CP09720 (identified by SERPA and MED tool), CP01850 (identified by MED tool), and CiuA (identified by TnFuZ system) presented the most promising results when applied as antigen in either diagnostic or vaccine formulation against the *C. pseudotuberculosis* infection. In addition, according to the results presented in these works (*Rezende et al., 2016*; *Silva et al., 2018*; *Silva et al., 2019*; *Brilhante Bezerra et al., 2020*) MED, SERPA, and TnFuZ are suitable tools to identify vaccine candidate and

diagnostic targets. However, more detailed studies are needed to assess the experimental efficacy of all identified targets in drugs, candidate vaccines, and diagnostic methods to combat the illness caused by this pathogen.

## Moonlight proteins in *C. pseudotuberculosis*

Moonlight proteins are multifunctional proteins, which exhibit different physiological functions in the same organism, depending on the cellular localization and multiple binding sites (*Jeffery, 1999*). These proteins are identified frequently in various types of organisms. Currently, hundreds of moonlighting proteins are present in databases like MoonProt (*Mani et al., 2015*) and MultitaskProtDB (*Franco-Serrano et al., 2018*). The world of moonlighting proteins comprises different proteins such as receptors, enzymes, transcriptional factors, chaperones, and scaffolds (*Jeffery, 2014*). In prokaryotic, moonlight proteins are present in the cell surface or extracellular environment. Many of these proteins act as adhesins when exposed to cell surfaces, thus contributing to bacterial colonization (*Jeffery, 2018*). However, the pathway used for these proteins to be secreted or exposed to the cell surface is unknown. Some studies suggest that the release of these proteins occurs through cell lysis, outer-membrane vesicles, increased membrane permeability, or decrease membrane integrity (*Ebner & Gotz, 2019*). In bacterial pathogen moonlight proteins play a role in the pathogenesis promoting adhesion or invasion to host cells and contributing to evasion of host immune response (*Henderson & Martin, 2013*; *Ebner & Gotz, 2019*).

Moonlight proteins have been detected in both exoproteome and surfaceome of *C. pseudotuberculosis* (Table 2). Interestingly, metabolic enzymes like enolase and GAPDH were not detected in the bacterial surfaceome grown in laboratory standard conditions, only in the shaving of RLC001_*ovis* collected directly from sheep lymph nodes (*Rees et al., 2015a*). In addition, enolase was also detected in the exoproteome of 1002_*ovis* upon serial passage to the murine host (*Silva et al., 2017a*). Studies showed that both enolase and GAPDH of *Streptococcus pneumoniae* contribute to adhesion and evasion from the host immune system (*Bergmann, Rohde & Hammerschmidt, 2004*; *Agarwal et al., 2012*). On the other hand, in *S. aureus*, GAPDH is related to metal uptake, invasion, and immunomodulation of the host immune system, while enolase contributes to invasion and biofilm formation (*Hemmadi & Biswas, 2020*). In the cell surface of *Mycobacterium tuberculosis*, GAPDH act as a lactoferrin receptor (*Malhotra et al., 2017*). Already enolase contributes to binds this pathogen to human plasminogen (*Rahi et al., 2017*). In addition to its metabolic functions and contributions to bacterial pathogenicity, some moonlight proteins, such as 1,6-bisphosphate aldolase (*Sun et al., 2015*), GAPDH (*Sun et al., 2017*), enolase (*Rahi et al., 2017*) and Hsp60 (*Bajzert et al., 2018*) are able to induce an immune response. Due to the results presented in these studies, these moonlight proteins might represent potential therapeutic targets. Nevertheless, studies futures are necessary to determine the extracytoplasmic function of these moonlight proteins on the pathophysiology of *C. pseudotuberculosis*.

**Table 2 Proteins described like moonlight protein detected in the exoproteome and surfaceome of *C. pseudotuberculosis*.**

| Protein name | Reference | *C. pseudotuberculosis* proteome | Canonical function and biological processes | Moonlight function | Microorganism | Reference |
|---|---|---|---|---|---|---|
| Superoxide dismutase [Mn] | *Silva et al. (2013a)* | Exoproteome | Antioxidant | Adhesion | *Mycobacterium avium* | *Reddy & Suleman (2004)* |
| 6-phosphogluconate dehydrogenase | *Silva et al. (2017a)* | Exoproteome | Pentose phosphate pathway | Adhesion | *Streptococcus pneumoniae* | *Daniely et al. (2006)* |
| Enolase | *Silva et al. (2013a), Rees et al. (2015a), Silva et al. (2017a)* | Exoproteome and Surfaceome | Glycolysis pathway | Plasminogen binding | *Aeromonas hydrophila* | *Sha et al. (2009)* |
| | | | | Plasminogen binding | *Bifidobacterium lactis* | *Candela et al. (2009)* |
| | | | | Plasminogen binding | *Borrelia burgdorferi* | *Floden, Watt & Brissette (2011)* |
| | | | | Binds plasminogen, fibronectin and epithelial cells | *Lactobacillus plantarum* | *Glenting et al. (2013)* |
| | | | | Plasminogen binding | *Neisseria meningitidis* | *Knaust et al. (2007)* |
| | | | | Plasminogen and laminin binds | *Staphylococcus aureus* | *Mölkänen et al. (2002), Carneiro et al. (2004)* |
| | | | | Plasminogen binding | *Streptococcus pneumoniae* | *Pancholi & Fischetti (1998)* |
| Glyceraldehyde-3-phosphate dehydrogenase (GAPDH) | *Pacheco et al. (2012), Silva et al. (2013a), Silva et al. (2013b), Rees et al. (2015a)* | Exoproteome and Surfaceome | Glycolysis pathway | Binds plasminogen, fibronectin, mucin and epithelial cells | *Lactobacillus plantarum* | *Glenting et al. (2013)* |
| | | | | Fibronectin binding protein-Plasminogen binding protein-Cell signaling kinase/ADP ribosylase-Neutrophil evasion protein | *Streptococcus pyogenes* | *Pancholi & Fischetti (1992)* |
| | | | | Plasminogen binding | *Bacillus anthracis* | *Matta, Agarwal & Bhatnagar (2010)* |
| | | | | Plasminogen binding protein / Adhesin for mucin | *Streptococcus pneumoniae* | *Bergmann, Rohde & Hammerschmidt (2004)* |
| Phosphoglycerate kinase | *Pacheco et al. (2011), Silva et al. (2013a)* | Exoproteome | Glycolysis pathway | Actin and plasminogen binding | *Streptococcus agalactiae* | *Boone & Tyrrell (2012)* |

| Protein name | Reference | *C. pseudotuberculosis* proteome | Canonical function and biological processes | Moonlight function | Microorganism | Reference |
|---|---|---|---|---|---|---|
| | | | | Plasminogen binding and complement inhibitor | *Streptococcus pneumoniae* | *Fulde et al. (2014)*; *Blom et al. (2014)* |
| | | | | Mucin binding | *Bifidobacterium longum* | *Nishiyama et al. (2020)* |
| Phosphoglyceromutase | *Pacheco et al. (2011)* | Exoproteome | Glycolysis pathway | Mucin binding | *Lactobacillus pentosus* | *Pérez Montoro et al. (2018)* |
| | | | | Plasminogen binding | *Mycoplasma pneumoniae* | *Gründel et al. (2015)* |
| Triosephosphate isomerase | *Pacheco et al. (2011)* | Exoproteome | Glycolysis pathway | Plasminogen binding | *Staphylococcus aureus* | *Ikeda & Furuya (2011)* |
| Chaperone GroEL | *Silva et al. (2013a)* | Exoproteome | Chaperone | Adherence-invasion | *Legionella pneumophila* | *Garduño, Garduño & Hoffman (1998)* |
| | | | | Adhesin | *Lactobacillus johnsonii* | *Kinoshita et al. (2016)* |
| | | | | Adhesin | *Clostridium difficile* | *Hennequin et al. (2001)* |
| Chaperone protein DnaK | *Pacheco et al. (2012)*, *Silva et al. (2013a)* | Exoproteome | Chaperone | Plasminogen binding | *Bifidobacterium lactis* | *Candela et al. (2010)* |
| | | | | Plasminogen binding | *Neisseria meningitidis* | *Knaust et al. (2007)* |
| | | | | Plasminogen binding | *Mycobacterium tuberculosis* | *Xolalpa et al. (2007)* |
| Peroxiredoxin | *Pacheco et al. (2011)*, *Silva et al. (2013a)* | Exoproteome | Antioxidant | Plasminogen binding | *Neisseria meningitidis* | *Aljannat et al. (2020)* |
| Elongation factor Tu | *Pacheco et al. (2011)*, *Silva et al. (2013a)* | Exoproteome | Translation elongation factor Tu | Attachment to human cells and mucins | *Lactobacillus johnsonii* | *Kinoshita et al. (2016)* |
| | | | | Fibronectin binding | *Mycoplasma pneumoniae* | *Widjaja et al. (2017)* |
| | | | | Mucin (MUC7)-binding protein | *Streptococcus gordonii* | *Kesimer et al. (2009)* |
| | | | | Binds mucin | *Bifidobacterium longum* | *Nishiyama et al. (2020)* |
| | | | | Receptor for host proteins | *Pseudomonas aeruginosa* | *Barbier et al. (2013)* |
| Chaperonin GroES | *Pacheco et al. (2011)*, *Silva et al. (2013a)* | Exoproteome | Chaperone | Macrophage adhesion | *Mycobacterium tuberculosis* | *Hickey et al. (2010)* |

## Hypothetical proteins

Bioinformatics tools have been helpful to obtain information about hypothetical proteins; these tools can predict domain and structure proteins, which provide information about

putative biological function or association to the biologicals pathway. *Araújo et al. (2020)* used several bioinformatics tools to perform the functional annotation of 80 hypothetical proteins from the core genome of 32 strains of *C. pseudotuberculosis*. In addition, due to the broad pipeline adopted in this study, the authors determined the physicochemical parameters and subcellular localization of these 80 hypothetical proteins.

A form of determining the evidence of hypothetical proteins is through genomic functional studies performing transcriptomics and proteomics studies. Some transcriptomic studies showed the gene expression of several CDS predicted as hypothetical proteins in the genome of *C. pseudotuberculosis* (*Pinto et al., 2014*; *Gomide et al., 2018a*; *Gomide et al., 2018b*; *Ibraim et al., 2019*; *Fu et al., 2020*). Regarding proteomics, this discipline takes advantage of transcriptomics studies over this type of study. Proteins are the final product of genic expression, and proteomes represent the complete set of proteins produced by a genome. Thus, a proteomic study can demonstrate the experimental evidence of hypothetical proteins, which from now on can be called unknown function proteins. According to proteomic studies of *C. pseudotuberculosis* the presence of unknown functional proteins range from 14 to 30% and many of these proteins were detected under different physiological conditions. Thus more efforts are necessary to characterize unknown function proteins and evaluate the role of these proteins in the *C. pseudotuberculosis* pathophysiology. Moreover, studies show that some unknown function proteins represent potential targets to be utilized in immunoassays (*Rezende et al., 2016*; *Silva et al., 2019*).

## CONCLUSION AND FUTURE DIRECTIONS

This review summarized the different proteomic strategies and bioinformatics utilized to promote a global functional analysis of the genome of *C. pseudotuberculosis* at the protein level. These studies allowed the validation of genomic *in silico* data, in addition to the identification of several proteins related to different biological processes, which contribute to the pathophysiology of *C. pseudotuberculosis*. Moreover, the inventory of proteins detected by different immunoproteomics strategies allowed the identification of promising targets to diagnostic methods, drugs, or vaccine formulations against the *C. pseudotuberculosis* infection. Overall, these proteomics studies contributed to our understanding of different aspects of the pathophysiology of *C. pseudotuberculosis*. Despite all this information generated, the incorporation of methodologies/techniques such as metabolomics, immunology, biochemical, and genetics is necessary to complement these proteomic studies.

### Funding

This work was supported by Brazilian Federal Agency for the Support and Evaluation of Graduate Education (CAPES), Minas Gerais Research Foundation (FAPEMIG), Paraense Amazon Foundation for Support to Studies and Research (FAPESPA), the

National Council for Scientific and Technical Research (CONICET), and the National Council for Scientific and Technological Development (CNPq). The funders had no role in study design, data collection and analysis, decision to publish, or preparation of the manuscript.

### Grant Disclosures
The following grant information was disclosed by the authors:
Brazilian Federal Agency for the Support and Evaluation of Graduate Education (CAPES).
Minas Gerais Research Foundation (FAPEMIG).
Paraense Amazon Foundation for Support to Studies and Research (FAPESPA).
National Council for Scientific and Technical Research (CONICET).
National Council for Scientific and Technological Development (CNPq).

### Competing Interests
Vasco Azevedo is an Academic Editor for PeerJ.

### Author Contributions
- Wanderson Marques da Silva conceived and designed the experiments, performed the experiments, analyzed the data, prepared figures and/or tables, authored or reviewed drafts of the paper, and approved the final draft.
- Nubia Seyffert performed the experiments, analyzed the data, prepared figures and/or tables, authored or reviewed drafts of the paper, and approved the final draft.
- Artur Silva conceived and designed the experiments, authored or reviewed drafts of the paper, and approved the final draft.
- Vasco Azevedo conceived and designed the experiments, authored or reviewed drafts of the paper, and approved the final draft.

### Data Availability
There was no raw data for this literature review.

### Supplemental Information
Supplemental information for this article can be found online at http://dx.doi.org/10.7717/peerj.12456#supplemental-information.

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
