# Peer review of "A journey through the Corynebacterium pseudotuberculosis proteome promotes insights into its functional genome"

_PeerJ, doi:10.7717/peerj.12456_

## Round 0.1 · original submission · Major Revisions

Dear Dr. da Silva and colleagues:

Thanks for submitting your manuscript to PeerJ. I have now received three independent reviews of your work, and as you will see, the reviewers raised some concerns about the review. I agree with these three reviewers, especially in regard to some missing references and other information that needs to be included in such a topical review.

There are many minor suggestions to improve the manuscript. Importantly, please ensure that an English expert has edited your revised manuscript for content and clarity. Please also consider adding the tables suggested by reviewer 2.

Therefore, I am recommending that you revise your manuscript, accordingly, taking into account all of the issues raised by the reviewers.

Good luck with your revision,

-joe

·

Basic reporting

This review paper is comprehensive with the stated topic with current information. This review have included all data compare and contrast with previous and current research related to the topic and the group of author is the expert subject matter. Well Done.

Experimental design

Well design manuscript and written and can be publish.

Validity of the findings

Yes all the data are compared and contrast with current and previous findings with subject matter of this review paper.

Additional comments

Well written and good studies have been conducted with the stated subject matter.

Reviewer 2 ·

Basic reporting

The authors are an established group studying the pathogenesis of C. paratuberculosis and have compiled a thoughtful review on the molecular basis of pathogenesis of this microbe. The review is very informative and exhaustive and the topic is not reviewed very often. The English language needs to be improved. Also there are some redundant information in the manuscript which needs to be trimmed to make the review more crisp and meaningful. The abstract needs to be rewritten highlighting the key thoughts from the review.

Experimental design

Specific comments
1.Lines 77-91 is not useful and should be removed.
Line 95 -what are these tools??
Line 97- rephrase. Does not make any sense; too long sentence.
Line 152- check grammar
Line 238-grammar
Line 240- bovine fetal? Please proof read extensively!
Line 260-isolated field???
Line 261- What is label based on demethylation? explain
Line 293-oxide nitric???
Lines 408-417 can be removed.

Validity of the findings

I would like a table showing the most relevant proteins which are identified across several studies and could be immunogenic and also could be vaccine candidates. Also, tabulate them according to function? Which ones do the authors think are best candidates? speculate.

Reviewer 3 ·

Basic reporting

The manuscript submitted by Wanderson Marques da Silva and co-author is summarizing the importance and progress of methodology in the field of functional –omics and how high-throughput techniques improved our understanding of C. pseudotuberculosis.

Experimental design

A check of the methology showed that several articles are neglected by the authors. An inclusion of manuscripts would imrove the manuscript.

Validity of the findings

The manuscript would benefit from a schematic overview, which approaches were used in the past and which strains were examined. Moreover, a comparison of the results from all different studies would help the reader. A short analysis of the common findings could specify targets and the present review would give the reader an outlook for future work.

Additional comments

General remarks
(i) Substantial English editing is recommended and several sentences need substantial rewording (see e.g.: line 100-104; 133-124; 123-124; 276-277; 298-301; 339-341; 405-406; 415-417; 419-420; 428-431; 434-435).
(ii) There are too many self-citations (48) in the manuscript, even when considering that the senior author contributed a lot in the field. Especially the introduction part should be revised, and more proper articles should be included. For example, Dorella et al., 2006a is not a good reference to describe the complex cell wall architecture of corynebacteria (line 46-47). A check of the methology showed that several articles are neglected by the authors. An inclusion of manuscripts would reduce the amount of self-citations.
(iii) Furthermore, the manuscript would benefit from a schematic overview, which approaches were used in the past and which strains were examined. Moreover, a comparison of the results from all different studies would help the reader. A short analysis of the common findings could specify targets and the present review would give the reader an outlook for future work.
(iv) In general, description of cited research is often rather superficial without giving any details of results obtained. See e.g.:
- Paragraph 77-91:
o What is “a high number”?
- Paragraph 130-147:
o how did growth in CDM improve the identiy? (line 140)
o how did the TPP method improve the results? (146-147)
o Is this “bias” confirmed?
- Paragraph 148-163:
o You started to compare gel-based and gel free proteomic approaches
o Could gel free approaches and bioinformatics techniques improve the quality of the results?
- Paragraph 164-184:
o 93 different proteins were detected in this study. Please specify, if this result was strain specific or in combination. Compare the findings and list them.
o Specify: the majority (line 174-175)
o You listed virulence factors (PLD, FagD and Cp40) identified in C231_ovis, are they present in 1002_ovis as well? You mention that this strain is described as less virulent. Please compare the virulence factors found in both strains. A Venn-diagram can visualize the differences and commonalities.
o Are both strains clinical isolates?
- Paragraph 210-226:
o Please list the 13 proteins from C. pseudotuberculosis and the 49 host proteins. This is a very interesting finding and should be explained in detail. Host-pathogen interaction networks can give a better understanding the infection process and gives hints for future research.
- Paragraph 227-237:
o For membrane proteomics it is recommended to use different proteases. Moreover, a filter aided sample preparation (FASP) an improve the identification. Please specify the protease
o Please list the 13 proteins from the bacteria, isolated from lymph nodes
o Please specify the identified 49 proteins from the host. This result shows the Host-pathogen interaction.
- Paragraph 257-264:
o Incorrect reference: this paragraph is about Rees et al., 2015b.
o Are there differences in the quality of the results between label-free and label-based methods and genomics?
- Paragraph 265-275:
o Please specify the differences between the strains, analyzed.
o A Venn-diagram or table would visualize the differences
- Paragraph 276-283:
o Rewrite the paragraph and focus on biofilm formation or show the differences between the different strains.
- Paragraph 321-329:
o Please explain the transposon based screening, this would also clarify the mutant strains
o How are the mice immunized? Are they immunized with the ciuA siderophore null strain and then challenged with a virulent strain?
o Strain differences?
o Is this paragraph necessary?
- Paragraph 330-345:
o What means “best immune response”? Please specify
o Leal et al., 2018 do not include experiments with rCP01850 (line: 341)
- Paragraph 346-362:
o Please list these 49 proteins with antigenic properties
o In line 361 you mentioned the resulting proteins. For consistency, please do this as well for the other proteins mentioned in your manuscript.
- Paragraph 363-373
o Again the proteins are not mentioned (line 367 and 372)
o Especially the potential targets would underline the importance of the use of proteomic strategies to evaluate the virulence potential.
- Paragraph 391-406:
o Please mention that the bacteria extracted from lymph nodes were compared to bacteria grown in liquid culture.
- Paragraph 407-417:
o Specify: pathogenic process specific (line 417)

Minor points
- Line 70: corynebacteria
- Line 97: “there is no in the literature a work with current information” plese revise
- Line 100 – 104: please rewrite the sentence.
- Line 123-124: please correct the sentence
- Line 165: What does MQD mean?
- Line 188: that alternative sigma factors are
- Line 191: sigmafactors
- Line 222: “bacterial surfaceome”
- Line 238: For the characterization of the surface proteome
- Line 238: VD57,
- Line 248 249: The whole-cell proteomic strategy allows the identification and quantification of thousands of proteins.
- Line 253 -254: to the identification of proteins
- Line 254: virulence
- Line 269: A DNA methylase
- Line 276 – 277: please rewrite the sentence
- Line 281: formation in prokaryotic organisms / prokaryotes
- Line 296: technologies
- Line 298 – 301: please correct the sentence
- Line 346: post genomic era
- Line 351-353: please rewrite the sentences
- Line 356: what means PSE
- Line 366: modelomic
- Line 367: strains
- Line 387: pathogen
- Line 390: response_(Henderson…)
- Line 402: binds
- Line 405 – 406: please rewrite the sentence
- Line 413: underscore after unveiled, please revise
- Line 414 – 417: please rewrite the sentence
- Line 419: proteins
- Line 441: identification
- Line 445 – 446: Future research, based on other
- Please revise the reference section.
o Do not start words with capital letters in the title
o DOI number is missing for some references
o Line 834: Streptococcus in italics
o Line 936 – 937: Corynebacterium pseudotuberculosis in italics
- I have found many spelling mistakes in the text. For this purpose, I suggest an English editing service or a native speaker.
- The word utilization has been used excessively in the text, please revise.
- NclassG+ is not available anymore, this should be mentioned or removed from the manuscript (line 162)
- Please add references for the CMNR group (line 44)

---

## Round 0.2 · Minor Revisions

Dear Dr. da Silva and colleagues:

Thanks for revising your manuscript. The reviewers are very satisfied with your revision (as am I). Great! However, there are some minor concerns still raised, and some edits to make. Please address these ASAP so we may move towards acceptance of your work.

-joe

Reviewer 2 ·

Basic reporting

The authors have answered most of the reviewer comments. I have no further queries

Experimental design

no comment

Validity of the findings

no comment

Reviewer 3 ·

Basic reporting

The mauscript entitled „A journey through the Corynebacterium pseudotuberculosis proteome promotes insights into its functional genome” is interesting and enjoyable to read. The article provides information about previous and current research and future challenges. The authors improved the manuscript significantly. Nevertheless, some minor changes are recommended.

Experimental design

In some titles of references, words always start with a capital letter. Please revise (Line: 654-656; 665-669; 672-674; 718-719; 720-721; 805-806; 852-854; 858-860; 865-868; 914-915; 918-920; 999-1000; 1021-1024; 1033-1036; 1096-1097).

Validity of the findings

Abbreviations of proteins should be written consistently. Always start with a capital letter, please revise the manuscript (Line: 289; 292; 303; 323; 324; 352; 462; 464).

Additional comments

The included tables and figures improve the manuscript considerably. However, the format of Table S.2 should be improved. It is hard to assign the “x” to the respective protein and reference. Moreover, it would be helpful to include the protein identifier. Especially to distinguish between the hypothetical proteins.

English editing:
Line 66: “described as major virulence factor this bacterium” the word “of” is missing
Line 152: produced
Line 198: CDM..
Line 199: ad
Line: 212-213: please revise the sentence
Line 223: proteins
Line: 237: proteomic
Line 344: “
Line 350: “differences quantitative” please revise
Line 387 – 388: please revise the sentence
Line 504 – 508: please revise the sentence
Line 513: an
Line 515 – 517: please rewrite the sentence
Line 521: “function biological” please revise
Line 532: “expression genic” please revise

---

## Round 0.3 · accepted · Accept

Dear Dr. da Silva and colleagues:

Thanks for revising your manuscript based on the concerns raised by the reviewer. I now believe that your manuscript is suitable for publication. Congratulations! I look forward to seeing this work in print, and I anticipate it being an important resource for groups studying Corynebacterium pseudotuberculosis genomics, biology and systematics. Thanks again for choosing PeerJ to publish such important work.

Best,

-joe